# Kinetic Study of 17α-Estradiol Activity in Comparison with 17β-Estradiol and 17α-Ethynylestradiol

**Tereza Bosakova [1,2], Antonin Tockstein [1], Zuzana Bosakova [1,*] and Katerina Komrskova [2,3,*]**

[1] Department of Analytical Chemistry, Faculty of Science, Charles University, Albertov 2030, 128 43 Prague 2, Czech Republic; tereza.bosakova@natur.cuni.cz (T.B.); atockstein@seznam.cz (A.T.)

[2] Laboratory of Reproductive Biology, Institute of Biotechnology, Czech Academy of Sciences, BIOCEV, Prumyslova 595, 252 50 Vestec, Czech Republic

[3] Department of Zoology, Faculty of Science, Charles University, Vinicna 7, 128 44 Prague 2, Czech Republic

\* Correspondence: bosakova@natur.cuni.cz (Z.B.); katerina.komrskova@ibt.cas.cz (K.K.);
Tel.: +420-2219-51231 (Z.B.); +420-6048-55871 (K.K.)

**Abstract:** 17α-estradiol (αE2), an endogenous stereoisomer of the hormone 17β-estradiol (E2), is capable of binding to estrogen receptors (ER). We aimed to mathematically describe, using experimental data, the possible interactions between αE2 and sperm ER during the process of sperm capacitation and to develop a kinetic model. The goal was to compare the suggested kinetic model with previously published results of ER interactions with E2 and 17α-ethynylestradiol (EE2). The HPLC-MS/MS method was developed to monitor the changes of αE2 concentration during capacitation. The calculated relative concentrations $B_t$ were used for kinetic analysis. Rate constants $k$ and molar ratio $n$ were optimized and used for the construction of theoretical $B(t)$ curves. Modifications in αE2–ER interactions were discovered during comparison with models for E2 and EE2. These new interactions displayed autocatalytic formation of an unstable adduct between the hormone and the cytoplasmic receptors. αE2 accumulates between the plasma membrane lipid bilayer with increasing potential, and when the critical level is reached, αE2 penetrates through the inner layer of the plasma membrane into the cytoplasm. It then rapidly reacts with the ER and creates an unstable adduct. The revealed dynamics of αE2–ER action may contribute to understanding tissue rejuvenation and the cancer-related physiology of αE2 signaling.

**Keywords:** 17α-estradiol; 17β-estradiol; 17α-ethynylestradiol; estrogen receptors; sperm; capacitation; HPLC MS/MS; kinetics; autocatalysis





## 1. Introduction

Estrogens are steroid hormones, and if present in the environment, they are classified as pollutants called endocrine-disrupting chemicals (EDCs) [1]. Even at extremely low concentrations (ng/L) they can have a detrimental impact on the endocrine systems of animals, including humans. EDCs can simulate the behavior of endogenous estrogens, which control several physiological processes, including male reproduction and in particularly sperm maturation [1–4].

An interaction between estrogens and their estrogen receptors (ER) has been characterized not only through nuclear receptors (nER) but also membrane (mER) and cytoplasmic (cER) receptors [5,6]. Several estrogenic hormones also lead to an increase in germ cell apoptosis and a decrease in sperm count [7,8]. It is therefore vital to study estrogens and ER interactions because they are heavily involved in the sperm maturation process called capacitation. It is during capacitation that sperm gain the capacity to fertilize the egg. We developed a novel approach utilizing the kinetic analysis of experimental data using theoretical modelling that resulted in new perspectives and interpretation of data.

Basic estrogens include the endogenous and biologically most active hormones 17β-estradiol (E2), 17α-estradiol (αE2) estriol (E3), estrone (E1) as well as synthetic 17α-

ethynylestradiol (EE2), used in hormonal contraceptives [9]. αE2 is the natural stereoisomer of E2 present endogenously. Its biological activity ranges between 1.5 and 5% of E2 activity with respect to species and tissue specificity [10–12]. It has recently been found to prolong the life of male mice, and to block inflammatory disfunction [10,13]. It has been discovered that αE2 is capable of binding to both ERα and ERβ receptors, both of which are present on the sperm plasma membrane (genomic effect). This is like the behavior of E2, but with much lower affinity [14]. In addition, αE2 can bind to cerebral ER-X receptors and membrane-associated plasma receptors (non-genomic effect) [15]. This can, for example, be manifested in a relaxation of the uterine smooth muscles [16]. αE2 has also been detected in small concentrations in samples of human urine and serum [17]. The responsiveness of male mice, but not castrated males or females, to αE2 treatment suggests sex specificity in the responses to αE2. This is caused by an interaction with male gonadal hormones, for example, the production of testosterone [18]. αE2 has a range of bioactive properties, including inflammatory and antioxidant effects [19] and an ability to inhibit the activity of $5\alpha$-reductase enzymes [20], which convert testosterone to dihydrotestosterone—a more potent activator of the androgen receptor. αE2 has been studied to demonstrate its benefit in the modulation of ovariectomy-mediated obesity and bone loss [21]. Treatment of male mice with αE2 has been shown to extend median lifespan [10,13,18,22]. It also improved male glucose tolerance during most of the adult life [23,24]. However, the mechanisms by which αE2 provides these benefits remain a matter of debate, even if αE2 elicits similar genomic binding and transcriptional activation through estrogen receptors to that of E2 [25].

The biological and physical processes that sperm undergoes after ejaculation are known as *capacitation* [26]. Only capacitated sperm gain progressive motility (sperm hyperactivation) and are capable of penetrating through the egg envelope (in a process known as the *acrosome reaction*) and to subsequently fertilize the egg [27]. Capacitation in vivo includes several fundamental processes: an increase in the liquidness of the cytoplasmic membrane; a reduction in cholesterol concentration in the membrane (termed cholesterol efflux); the opening of ion channels followed by a change in membrane potential; the loss of some surface proteins (and phosphorylation of proteins leading to hyperactivation); and the *acrosome reaction* [28]. It is significant that EDCs can affect all these physiological processes and modify their interactions [4].

To study capacitation, the process needs to be simulated in vitro and must fulfil strict requirements regarding sperm incubation. This includes a capacitation/fertilization medium containing albumin and calcium ions at a temperature of 37 °C, maintained in an atmosphere with 5% $CO_2$. These specific conditions imitate the physiological environment in the female genital tract [29]. To study how the presence of estrogens effect sperm capacitation in vitro, it is important to determine the capability of sperm to bind these hormones and the dynamics of this process. By using kinetic analysis on experimental data, we may be able to resolve this process. A similar approach has already been used to monitor time-dependent changes in mRNA concentration [30,31].

The developed HPLC-MS/MS methods enabled monitoring of the concentration changes of the free, unbound hormone (E2 or EE2) during time-dependent capacitation of mouse sperm (dose concentrations 200, 20 and 2 μg/L). Then, this was achieved through the dependence of the relative concentration of the free hormone (*B*) on capacitation time (*t*) [32,33]. It was apparent from the shape of the *B*(*t*) curves, obtained for previously studied E2 or EE2, that the hormones form unstable adducts with the receptors and that the formation of these adducts acts autocatalytically [32,33]. It was therefore hypothesized that analogous behavior could also be occurring for the interaction between αE2 and ER in maturing sperm.

This research was performed to determine whether the action of αE2 during the capacitation of mouse sperm exhibits a similar pattern to that of E2 and EE2 [32,33]. Firstly, to answer this question, we developed an HPLC-MS/MS method capable of monitoring the concentration changes of unbound free αE2 during the time-dependent capacitation of

mouse sperm (in a capacitation M2 medium). Secondly, we subjected the obtained results to kinetic analysis. To make a valid comparison between αE2 and previously measured hormones E2 and EE2, the following parameters were kept constant: measurement methodology; concentrations of hormone; evaluation of the obtained data for kinetic analysis; and mathematical procedure.

## 2. Results

### 2.1. Monitoring the αE2 Concentration during Time-Depedent Capacitation Using the HPLC-MS/MS Method

Similar to a previous study of the concentration changes in the estrogenic hormones E2 [32] and EE2 [33] during the time-dependent capacitation of mouse sperm, the HPLC-MS/MS method was selected for an analogous study of the estrogenic hormone αE2. As a first step, under optimized separation and capacitation conditions (see Chapter 4.2), the concentration of free unbound αE2 (dependent on the capacitation time) was monitored for a dose concentration of 200 μg/L. The resultant data are shown in Figure 1. It is apparent from these data that sample values first decreased to a capacitation time of 150 min, where they attained a minimum, and thereafter increased. In contrast, the blank values remained essentially constant during the whole capacitation time. For dose concentrations of 20 and 2 μg/L, the samples exhibited practically no difference compared to the blank and almost constant values that were obtained for all the monitored capacitation times, varying in intervals from 17.8 to18.6 μg/L for dose concentrations of 20 μg/L and from 1.7 to 1.8 μg/L for the dose concentration of 2 μg/L.

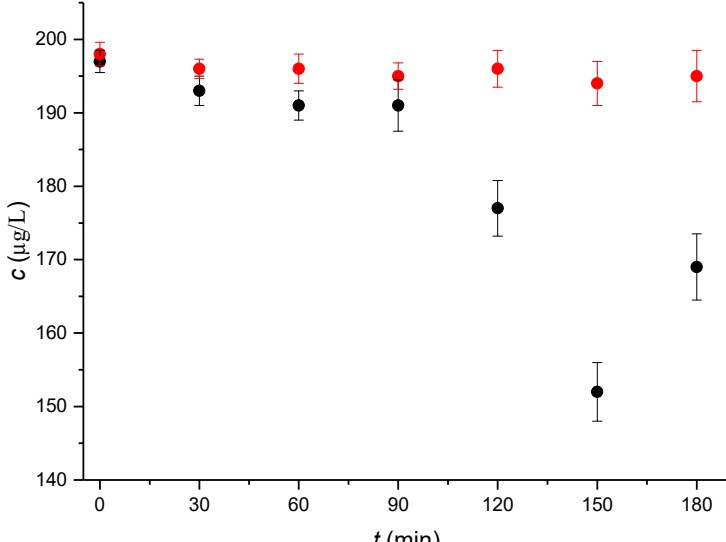

**Figure 1.** Dependences of the concentration of total unbound αE2 on the time of mouse sperm capacitation in vitro. The tested concentration of αE2 in capacitating medium was 200 μg/L. The samples were prepared in three parallel sets, where each set represented sperm collected from one individual (black dots). The blanks were prepared and measured in parallel (red dots). Experimental conditions: 50/50 (*v/v*) ACN/$H_2O$, both containing 0.1% HCOOH, were measured in the multiple reaction monitoring (MRM) mode for transition 255.0→158.9, and the error bars were calculated using the standard deviations ($n' = 3$) according to the methodology used for E2, EE2 hormones [32,33].

The measured values for a dose concentration of 200 μg/L yielded the values of the relative concentration $B_t$ defined as $C(t)/C(t = 0)$ for the individual capacitation times. The relevant values are given in Table 1.

**Table 1.** Relative concentrations ($B_t$) calculated from the means of measured $\alpha$E2 time-dependent concentrations $C$ ($n' = 3$) obtained during capacitation for the tested concentration of 200 µg/L $\alpha$E2 (see Figure 1); mean $\pm$ standard error of the mean.

| Time (min) | 0 | 30 | 60 | 90 | 120 | 150 | 180 |
|---|---|---|---|---|---|---|---|
| $B_t$ | $1.000 \pm 0.002$ | $0.985 \pm 0.002$ | $0.981 \pm 0.003$ | $0.979 \pm 0.008$ | $0.899 \pm 0.010$ | $0.772 \pm 0.012$ | $0.863 \pm 0.013$ |

The $B_t$ values of the samples and blank obtained for the dose concentration of 20 µg/L oscillated around 0.957 and for the dose concentration of 2 µg/L around 0.944. Due to their constant values throughout the capacitation time, they were not involved in further kinetic studies.

### 2.2. Kinetic Analysis of the HPLC-MS/MS Data

If a curve is fitted through the experimentally obtained values of $B_t$, it might at first appear that this curve has a similar shape to the one obtained from the experimentally obtained $B_t$ values of the previously studied E2 and EE2 hormones. In our previous research targeting the E2 and EE2 reaction with ER in mouse sperm during sperm capacitation, it was found that both E2 and EE2 display a common characteristic in the autocatalytic formation of an unstable adduct between the relevant hormone (E2 or EE2) and the estrogen receptors (cER) located in the cytoplasm, and this was manifested by a decrease followed by an increase in the $B_t$ value [32,33].

Based on the published data for EE2 [33], the decomposition of the adduct could be described by two hypothetical possibilities that could also apply for $\alpha$E2: (a) decomposition produces active estrogen hormone and active R (cER) receptor, which are both capable of further interactions; (b) the decomposition produces active estrogen hormone and inactive receptor R′, incapable of further interaction. Because both components formed in the first reaction (a) are hypothetically active, it is simple to imagine a reverse (equilibrium) reaction between the hormone and the estrogen receptor R in the cytoplasm. On the other hand, the second reaction (b) is a unidirectional leading to a deactivation of the receptor. These two reactions (a) and (b) could take place simultaneously and this phenomenon can be considered as a disturbance of the equilibrium (i.e., a pseudo-equilibrium reaction). The process observed for $\alpha$E2 together with the set of equations can be described by the following simple kinetic scheme (see Figure 2).

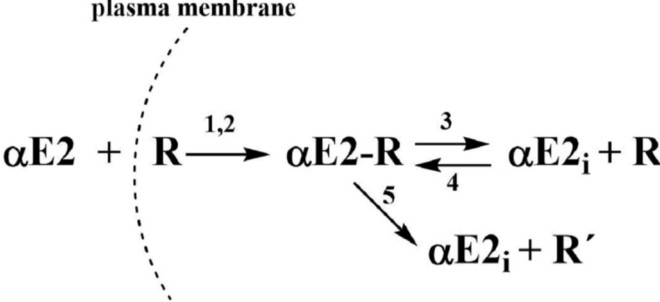

**Figure 2.** Kinetic scheme where: $\alpha$E2 is the extracellular hormone; $R$ is the active sperm receptor; $\alpha$E2-R is the adduct; $\alpha$E2$_i$ is the internal free hormone and $R'$ is the inactive sperm receptor. The plasma membrane is not a flat surface but is an intermediate phase with an external orientation facing the hormone solution, and an internal orientation facing the cytoplasm. 1 and 2 represent rate constants of non-autocatalytic (1) and autocatalytic (2) formation of $\alpha$E2-R, ongoing simultaneously; 3 and 5 represent rate constants of adduct decomposition, and 4 represents a rate constant of reverse reaction.

Based on the scheme in Figure 2, the monitoring of $\alpha$E2 concentration by HPLC/MS-MS is related to the overall concentration of free, unbound hormone, i.e., the original $\alpha$E2 and the internal $\alpha$E2$_i$. After centrifugation, both components of the hormone were present

in the supernatant. However, the hormone bound in the complex with the receptor (adduct: αE2-R) remains in the sediment and its concentration was not measured.

Additionally, the kinetic scheme employs kinetic products and rate constants in the form of differential equations. For brevity, the term αE2 is replaced in the equations by the symbol *E* (also designating its molar concentration under various experimental conditions), *R* designates the cytoplasmic estrogen receptor (cER) and its concentration, symbol (*ER*) designates the adduct, $E_i$ is the internal free hormone in the cytoplasmic membrane, and $R'$ is the inactive cER receptor.

$$\frac{-\mathrm{d}E}{\mathrm{d}t} = E \times R(K_1 + K_2 \times (1 - \varepsilon)) \tag{1}$$

$$\frac{-\mathrm{d}R}{\mathrm{d}t} = E \times R(K_1 + K_2 \times (1 - \varepsilon)) + K_4 \times E_i \times R - K_3 \times (ER) \tag{2}$$

$$\frac{\mathrm{d}(ER)}{\mathrm{d}t} = E \times R(K_1 + K_2 \times (1 - \varepsilon)) + K_4 \times E_i \times R - (K_3 + K_5) \times (ER) \tag{3}$$

Equation (1) describes the decrease in the hormone *E* by a reaction with the receptor *R*. Equation (2) describes the decrease in the receptor through its reaction with the external and internal pool of the hormone and the increase by the subsequent adduct decomposition. Equation (3) describes the increase in the adduct through the autocatalytic reaction and reverse reaction including the decrease through the equilibrium ($K_3$) and pseudo-equilibrium ($K_5$) reactions.

These differential equations were subsequently solved by the Runge–Kutta method, used to determine the values of the parameters of the theoretical *B*(*t*) curve, i.e., the rate constants $K_1$–$K_5$ and the value of the molar ratio of the reacting components, parameter *n*. These values were optimized by looking for the minima in the absolute values of the difference between the theoretical and experimental $B_t$ values using the MATLAB program. The results of optimization of the rate constants and parameter *n* for a concentration of 200 μg/L are listed in Table 2. The derivation of the individual differential equations and reasons for their specific position are described in Reference [33].

**Table 2.** Calculated constants obtained for an αE2 concentration of 200 μg/L, where $K_1$–$K_5$ are the overall rate constants and *n* is the molar ratio.

| $K_1$ | $K_2$ | $K_3$ | $K_4$ | $K_5$ | *n* |
|-------|-------|-------|-------|-------|-----|
| 0.01 | 4.0 | 5.0 | 0 | 0 | 0.01 |

The *B*(*t*) curves formed using constants $K_1$–$K_5$ for the various values of the parameter *n* are given in Figure 3. The way in which the shape and position of the minima of the *B*(*t*) curves depend on the value *n* is apparent from the shapes of the individual curves. It is evident that the fitting of the experimentally obtained points (black points) with the minimum of the calculated *B*(*t*) curve is tighter for extremely small *n* values (red or green points).

As mentioned above, the shape and form of the *B*(*t*) curve for αE2 might initially appear to agree with the *B*(*t*) curve (obtained for the reaction of the previously studied EE2 hormone [33]), corresponding to the autocatalytic formation of an unstable adduct. However, important differences in the results necessitate the development of a different reaction mechanism. The most significant differences between the αE2 and EE2 data are as follows:

(i)   The initial branch of the hypothetical curve that would be obtained by fitting the experimental points is the opposite to that of the autocatalytic curve.

(ii)  The slope of the tangent to this hypothetical curve decreases in its initial region to zero with time.

(iii) The determined value of *n* is too low for a dose hormone concentration of 200 μg/L.

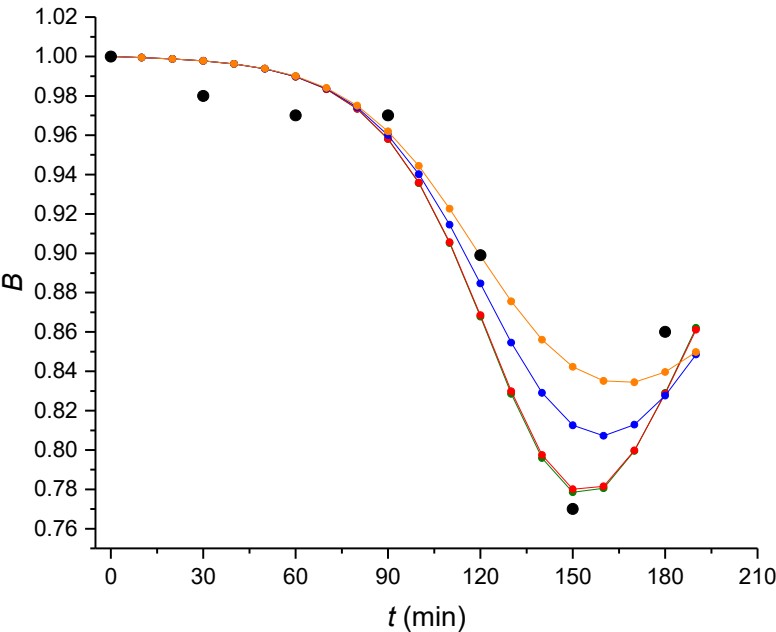

**Figure 3.** Shape of the theoretical $B(t)$ curves calculated with optimized $K_1$–$K_5$ constants (see Table 2) for various values of the molar ratio $n$: 0.01—green line; 0.1—red line; 1—blue line; 2—orange line; experimentally obtained $B_t$ values—black dots.

As a result, further optimization of the rate constants was performed to enable a better fitting of the experimental points obtained during a capacitation time of 0–90 min, and the resultant data can be found in Table 3.

**Table 3.** Calculated constants obtained for an αE2 concentration of 200 μg/L, where $K_1$–$K_5$ are the overall rate constants and $n$ is the molar ratio.

| $K_1$ | $K_2$ | $K_3$ | $K_4$ | $K_5$ | $n$ |
|-------|-------|-------|-------|-------|-----|
| 7 | 10 | 49 | 12 | 20 | 0.01 |

Figure 4 gives the experimentally determined values of $B_t$ (black dots) together with the curves obtained using the optimized kinetic constants given in Table 2 (Curve 1) and Table 3 (Curve 2). The initial branch of Curve 2 up to the intersection with Curve 1 (at 90 min of capacitation), corresponds to the difficulty of the αE2 hormone finding an entrance into the external facing surface of the plasma membrane. αE2 does not initially pass through the inner facing layer of the plasma membrane and it accumulates with increasing strain between the plasma membrane lipid bilayer. When the amount of αE2 in the plasma membrane reaches a critical value, the inner layer of the membrane becomes permeable for αE2. This results in the internalization of αE2 within the cytoplasm, where the hormone rapidly reacts autocatalytically with the estrogen receptor (cER) and forms an unstable adduct (note the sharply decreasing branch of Curve 1), which subsequently decomposes (see the final, increasing branch of Curve 1). The described process results in a $B(t)$ angled curve. Because the critical amount of αE2 is extremely low relative to the total content of cER in the cytoplasm, the value of the parameter $n$ is extremely low in the reaction of the hormone with the cytoplasm receptors.

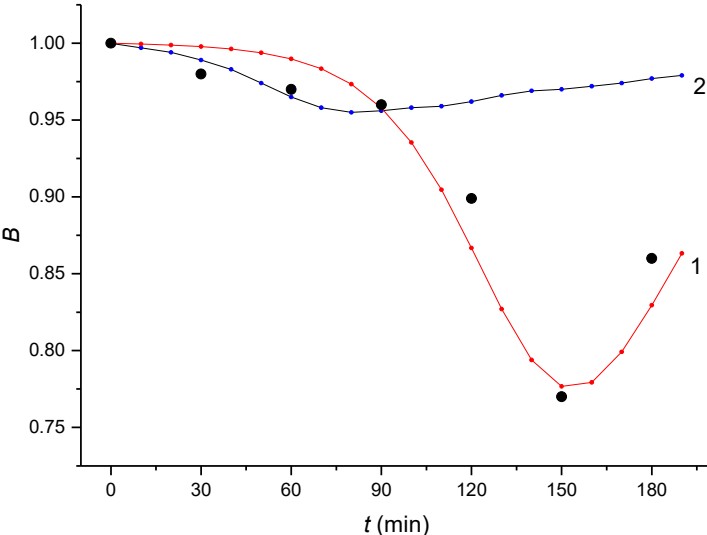

**Figure 4.** The values of the relative concentration $B_t$ obtained experimentally by the HPLC/MS/MS method (black dots) together with the theoretically calculated $B(t)$ curves obtained by integration of the kinetic equations with optimized constants given in Table 2 (Curve 1) and Table 3 (Curve 2). The dose concentration of αE2 was 200 μg/L.

The initial branch of Curve 1 should be much slower for a concentration of 20 μg/L (or 2 μg/L) compared to a concentration of 200 μg/L, so the minimum should occur after 180 min. Significantly, this is after the standard capacitation time. This theory is confirmed by the experimental findings.

## 3. Discussion

From the kinetic analysis of experimentally determined values of the relative concentration ($B_t$) for individual capacitation times, the results show that one uniform process resulted in the formation of an adduct formed by the hormones (E2, EE2, αE2) and the sperm cytoplasmic estrogen receptor (cER). The formed adducts are unstable and decompose to the original free hormones (E2, EE2, αE2) and cER [32,33]. If the decomposition results in an unaltered receptor (that can react with the internal pool of the hormone to form the adduct by a reverse reaction), then this is defined as an equilibrium reaction. If an inactive receptor is present, then the equilibrium reaction is interrupted and leads to a pseudo-equilibrium reaction.

The formation of the adduct is an autocatalytic reaction characterized by the fact that the tangent to the $B(t)$ curve becomes steeper during the reaction. The autocatalysis consists of the formed adduct that relaxes the plasma membrane and facilitates a passage of the hormones (E2, EE2, αE2) with increased production of the adduct. This process is then repeated. Based on the outcome of kinetic analysis, the difference in the response of the individual hormones appears to be due to a different mode of transport of the hormones through the plasma membrane.

E2 is an endogenic hormone that requires defined membrane estrogen receptors to generate the required response. The number of receptors on the surface of sperm is determined by the adsorption isotherm. E2 can only penetrate through the plasma membrane to form the adduct through the receptors. Because the isotherm is proportional to the concentration of the external E2 hormone; the formation of the adduct is also dependent on the same process. The minimum value on the time axis is an important element on the $B(t)$ curves and is dependent on the dose hormone concentration. Specifically, the minimum shifts to later capacitation times as the dose hormone concentration decreases [32]. This result clearly indicates that an endogenic hormone is involved. The depth of the minimum on the $B(t)$ curve deepens as the value of $n$ (molar ratio) decreases. The ratio of the two neighboring $n_1/n_2$ values obtained for two neighboring dose concentrations ((200;

20 μg/L) or (20; 2 μg/L)), equals 1/10, and this is in accordance with the ratio of two neighboring dose concentrations. It represents another typical characteristic behavior of the E2 hormone [32]. However, the described kinetic action for the E2 hormone was not the case for the other estrogen hormones.

The experimentally determined $n$ values for the individual dose concentrations are much smaller for the EE2 than the corresponding $n$ values for E2. The EE2 hormone does not seem to require a specific membrane estrogen receptor to pass through the sperm plasma membrane and therefore this reaction is rapid. A characteristic feature of the EE2 action is the fact that the time coordinates of the minimum are practically constant and, therefore, they are independent of the hormone dose concentration [33].

The passage through the plasma membrane is substantially more complicated for the αE2 hormone. Firstly, αE2 does not appear to pass through the inner facing layer of the plasma membrane and it accumulates between the plasma membrane lipid bilayer. It is manifested by the relatively long, flat part of the $B(t)$ curve. When the amount of αE2 within the plasma membrane reaches a critical value, the inner layer of the membrane becomes permeable for αE2, followed by internalization of αE2 within the cytoplasm. Here, the hormone rapidly reacts autocatalytically with cytoplasmic receptors and forms an unstable adduct, demonstrated by a sharp decrease in the $B(t)$ curve to the minimum point. The critical amount of this hormone is small compared to the overall amount of estrogen receptors in the sperm cytoplasm. The value of parameter $n$ is also extremely low in the reaction of the hormone with the cER. It is important to note that the kinetic model suggested for αE2 (a hormone with low hormonal activity) is successfully described by the same differential equations as EE2 that represent a synthetic hormone. These conclusions support the validity of the proposed kinetic models. Suggested mathematical models, validated by experimental data, allowed for the development of hypotheses for possible interactions between estrogen hormones and estrogen receptors using the sperm capacitation model.

αE2 has also been discovered to bind to brain-expressed ER-x [15] and breast-expressed ER-X [34] membrane-associated receptors to mediate estrogen-specific signaling and classical ER [6]. These receptors have a sequence homology to ER which is known to be expressed in gonads including sperm. However, the role of these receptors in male reproduction remains to be determined [6,35]. It is of importance that αE2 was described to be biologically active in human breast cancers [11]. Since testicular germ cell cancer (TGCT) is associated with polymorphism in estrogen receptors and steroid metabolism genes, sex steroid action is regarded as involved in the development of TGCT [36]. As estrone (E1) is metabolized to estradiol (E2), there is a strong correlation with functional activity, estradiol level, or estrogen-related outcomes between TGCT [37] and breast cancer [11]. αE2 represents the endogenous estradiol isomer for which biological roles are described but not fully understood. αE2 is shown to play a crucial role in regenerating brain tissue after injury [38,39], and its significant role in cancer-related events including TGCT has been published. The novel understanding of the dynamics of αE2–ER crosstalk on the sperm model presented in this paper has the potential to be used for advanced understanding of the physiology of its signaling action within cells.

## 4. Materials and Methods

In order to guarantee the comparison between studied hormones, relevant materials and methods were used according to previously published kinetic studies [32,33]. The Materials and Methods section was updated to include αE2.

### 4.1. Chemicals, Reagents and Animals

Acetonitrile (ACN) for LC-MS, Chromasolv (purity ≥ 99.9%), deuterated β-estradiol-16,16,17-d3 (estradiol-d3) (purity 98%), and commercial capacitating M2 culture media for in vitro sperm capacitation and fertilization (M7167) were purchased from Sigma-Aldrich (Steinheim, Germany). Ethanol (purity ≥ 96% p.a.) was obtained from Lachner (Neratovice,

Czech Republic). Paraffin oil was provided by Carl Roth (Karlsruhe, Germany). Formic acid (HCOOH) (LC-MS LiChropur, purity 97.5–98.5%), 17α-estradiol (purity 98%) and deionized water (for UHPLC-MS LiChrosolv) were obtained from Merck (Darmstadt, Germany).

The laboratory inbred house mouse strain BALB/c was used for the experiments. The mice were housed in the animal facilities of the Institute of Molecular Genetics of the Czech Academy of Science, Prague. Food and water were supplied ad libitum. All the animal procedures and all the experimental protocols were approved by the Animal Welfare Committee of the Czech Academy of Sciences (Animal Ethics Number 66866/2015-MZE-17214, 18 December 2015).

### 4.2. Instrumentation and Chromatographic Conditions

The HPLC equipment (Agilent Technologies, Waldbronn, Germany) consisted of 1290 Infinity Series LC (a quaternary pump, degasser, thermostatic auto sampler and column oven). A Triple Quad LC/MS 6460 tandem mass spectrometer (Agilent Technologies, Waldbronn, Germany) with an electrospray ionization interface was used for the detection. The signal was processed, and the data were retreated using the MassHunter Workstation Acquisition and MassHunter Qualitative Analysis Software (Agilent Technologies, Waldbronn, Germany).

All the instrumental MS-MS parameters were optimized. The ESI (+) conditions in the MRM mode for αE2 were capillary voltage 4000 V, nebulizer pressure 55 psi, gas temperature 350 °C, and a nitrogen flow rate of 10 L/min.

For αE2, the *m/z* 255.0→158.9 and for estradiol-d3, the *m/z* 258.5→158.9 transitions were monitored (fragmentor voltage 130 V, collision energy 14 V, parameter dwell 200 ms).

The separation system was based on publications [33,40] with a Kinetex EVO column C18 (100 × 3.0 mm, 2.6 μm, Phenomenex, Torrance, CA, USA) and a mobile phase containing a binary mixture of ACN/water with the addition of 0.1% HCOOH in both parts at a volume ratio of 50/50 (*v/v*); the flow rate was 0.3 mL/min. The column temperature was held at 21 ± 0.5 °C. The amount of sample injected equaled 7.5 μL. Estradiol-d3 was added to each sample as an internal standard (IS) at a final concentration of 25 μg/L (diluted in the capacitating medium). The retention times of IS and αE2 were 3.0 and 3.4 min, respectively. Because of the complex capacitating medium containing inorganic and organic components, of which especially bovine serum albumin (4.0 g/L) can cause difficulties during the separation and detection processes, the eluate was fed to waste from 0 to 2.5 min and to the MS detector only from 2.5 to 5 min.

The linearity of the method was determined from the calibration curve constructed by plotting the ratio of the peak areas of αE2 to that of IS against the analyte concentration. It was statistically analyzed by 1/x weighted linear regression analysis using the least-squares regression method. The obtained data were linear (y = 0.0544x + 0.0702, $R^2$ = 0.9987) in the whole measured calibration range 1–250 μg/L. Relative standard deviations (*n* = 5) varied from 1.8–5.9%. The limit of detection (0.52 μg/L) and limit of quantitation (1.49 μg/L) were calculated as the 3.3 × σ/S and 10 × σ/S ratios, respectively, where σ was the baseline noise of the sample and *S* was the slope of the regression curve (based on the ratio of the peak heights of αE2 to that of IS against the analyte concentration) constructed from the calibration curve.

### 4.3. Sample Preparation

35 mM culture dishes purchased from Corning (Corning, NY, USA) were used for the capacitation in vitro. The Olympus CX 21 inverted-light microscope and Olympus epifluorescent microscope were supplied by Olympus (Prague, Czech Republic). The NB-203 incubator was purchased from N-BIOTEK (Gyeonggi-do, Korea). The Telstar Bio-IIA incubator and BioTek laminar box from N-BIOTEK (Gyeonggi-do, Korea) were used for the in vitro sperm cultivation.

First, a stock solution of the αE2 standard with a concentration of 200 mg/L was prepared in ethanol, from which the working ethanol solutions of an αE2 concentration of

20 and 2 mg/L were diluted. In the laminar box, 10 μL of the working ethanol solution of αE2 with the appropriate concentration (200, 20 or 2 mg/L) was pipetted and diluted with capacitating medium to a volume of 10 mL in the test tube, so that the final test concentration of αE2 in the capacitating medium was obtained (200, 20 or 2 μg/L) and the ethanol content was minimized and remained constant. Subsequently, 100 μL of solution with the appropriate concentration was pipetted into the fertilization Petri dish. The pipetted mixture in the Petri dish was covered with 1 mL of paraffin oil. The prepared Petri dishes were placed in the incubator and tempered for 60 min at a temperature of 37 °C and with 5% $CO_2$ in the air.

The spermatozoa which were recovered from the distal region of the *cauda epididymidis* were placed in fertilization Petri dishes with a capacitating medium and paraffin oil and then placed for 10 min in the incubator for sperm release. Next, the stock concentration of mouse sperm was adjusted to $5 \times 10^6$ sperm/mL. The control experiments monitoring sperm motility and viability were run in parallel and controlled under a microscope. No toxicological effect of αE2 in capacitation media was found.

The biological sample was prepared as follows: following 60 min of tempering, 5 μL of the stock sperm was added to 100 μL of the αE2 sample in capacitating medium (200, 20 or 2 μg/L). For each capacitation time (0–180 min), 8 Petri dishes containing 105 μL of sample covered with 1 mL of paraffin oil were prepared. The dishes prepared in this way were incubated again under the same conditions for various time periods (0, 30, 60, 90, 120, 150 and 180 min after adding the sperm), during which sperm capacitation took place. After the individual times, samples (only the solution with the capacitating medium, without the paraffin oil) were pipetted from all 8 Petri dishes into a single micro-test tube, which was centrifuged for 10 min at 12,000 rpm. In this way, the sperm were separated from the solution and approximately 600 μL of supernatant was obtained for HPLC-MS/MS analysis of free, sperm unbound αE2. This sample represented one sampling time during capacitation.

In order to eliminate any systematic errors during sample preparation (partial evaporation of samples during incubation, differences in collection of supernatants after centrifugation, etc.), reference samples (blanks) without the addition of mouse sperm were prepared simultaneously under the same experimental conditions.

Prior to the actual HPLC-MS/MS analysis, 20 μL of IS (estradiol-d3) with a concentration of 250 μg/L was added to each biological sample as well as to a blank (180 μL).

The matrix effect was investigated by a comparison of results (αE2/IS peak area ratios) obtained for samples prepared by two different procedures: (i) the sample was prepared by the addition of αE2 and IS to the supernatant, obtained after the capacitating medium with sperm covered with paraffin oil was tempered, the paraffin oil removed and capacitating medium centrifuged; (ii) the sample was prepared by addition of αE2 and IS to the capacitating medium. Each experiment was performed in triplicate for concentrations of 200, 20 and 2 μg/L. The recovery of the samples with αE2 was 96.8% for 200 μg/L, 98.5% for 20 μg/L and 91.3% for 2 μg/L.

**Author Contributions:** T.B. performed sperm capacitation, the HPLC-MS/MS experiments and the relevant statistics; T.B., Z.B. and A.T. designed and performed the kinetic analysis part and wrote the relevant part of the manuscript; K.K. designed and supervised the biological part of the project and wrote the relevant parts of the manuscript. All the authors reviewed the manuscript. All authors have read and agreed to the published version of the manuscript.

**Funding:** This research was funded by the Grant Agency of the Charles University GAUK 693118 and SVV260560, by the Grant Agency of the Czech Republic No. GA-20-20217J, by the project "BIOCEV–Biotechnology and Biomedicine Centre of the Academy of Sciences and Charles University" (CZ.1.05/1.1.00/02.0109) and from the European Regional Development Fund (www.biocev.eu), and by the Institutional support of the Institute of Biotechnology RVO: 86652036.

**Data Availability Statement:** Data sharing not applicable.

**Conflicts of Interest:** The authors declare no conflict of interest.

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
