# Peer review of "Kinetic Study of 17α-Estradiol Activity in Comparison with 17β-Estradiol and 17α-Ethynylestradiol"

_catalysts, doi:10.3390/catal11050634_

Round 1
Reviewer 1 Report
In this reviewer opinion, it is necessary when submitting a manuscript to attach also a word document with the line numbers, to allow the interaction between reviewers and Authors in an easier way.
Comments:
-Please change the manuscript title. Avoid the use of questions, standing out the result and not the aim.
-English language should be revised.
Abstract:
-“in interaction in question take place…” remove “,”
-Sperm maturation is NOT the same than sperm capacitation. Sperm capacitation is the acquisition of the fertilizing ability, while sperm maturation starts before.
-Please add some lines at the end regarding the implications of the findings, or a conclusion.
Introduction
-Where is alpha-E2 naturally found in vivo? Follicular fluid? Oviductal fluid? Seminal fluid? None of them?
-Please, add some lines justifying the selection of the concentrations used in this study.
Results
-Similarly to the… please remove the “,”
-Where toxicological experiments done? Which are the implications of this hormone supplementation in the sperm physiology?
Discussion
With all my respects, discussion section is very poor. Please, add some lines regarding the importance of the study, where it can be applied/found, which are the implications of the findings, …
This is necessary for the acceptation of this manuscript for publication.
Bibliography is also poor (only 26 references), and old manuscript were cited. Please, update the references and add the biological implication of this study.
These modifications should be carried out to accept the manuscript. Unfortunately and in this reviewers opinion, the manuscript is not acceptable in its present form.
Reviewer 2 Report
Review of the manuscript:
"Do estrogens act the same during sperm capacitation? Kinetic study of 17α-estradiol activity in comparison with 17β-estradiol and 17α-ethynylestradiol"
by T. Bosakova et al.
The authors present a theoretical kinetic model of the αE2 and sperm ER interactions and use experimental data to validate it. They also provide an interesting interpretation of the model obtained.
The manuscript is weel written and well organized. The material is quite interesting and the result are suggestive of an intriguing biological mechanisms.
A general discussion about time series kinetics is missing from physiological and molecular data. Some general references and reviews should be added. I suggest the following:
Palumbo MC, Farina L, Paci P. Kinetics effects and modeling of mRNA turnover. Wiley Interdiscip Rev RNA. 2015 May-Jun;6(3):327-36. doi: 10.1002/wrna.1277. Epub 2015 Mar 1. PMID: 25727049.
A robustness analysis should be included to evaluate the effect of some variability on the parameters of the model.
Round 2
Reviewer 1 Report
I appreciate the modifications made by the Authors, which in this Reviewers opion helped to improve the manuscript quality. However, manuscript reading remains slow and difficult. I will recommend to revise the whole manuscript for English language prior to resubmit it, favouring an easy reading.
Some comments:
-Introduction L47-52. Please rewrite, add dots when needed and make it clear.
L62: alpha-E2 has been studied / is studied.
L77. "it is disturbing" not too appropiated. I suggest "it is intringuing..." or something similar.
L100. Okay for this paragraph, but please add somewhere the hormone concentrations.
L315."allowed the development of..."
In general, please revise the paragraphs that were recently added.
